# Seasonal Variation in Chemical Compositions of Essential Oils Extracted from Lavandin Flowers in the Yun-Gui Plateau of China

**DOI:** 10.3390/molecules26185639

**Published:** 2021-09-17

**Authors:** Zhenni Liao, Qing Huang, Qiming Cheng, Sardar Khan, Xiaoying Yu

**Affiliations:** 1Key Laboratory of Urban Environment and Health, Institute of Urban Environment, Chinese Academy of Sciences, Xiamen 361021, China; 2018010157@m.scnu.edu.cn (Z.L.); chengren008@163.com (Q.C.); sardar.khan2008@yahoo.com (S.K.); 2Chenzhou Institute of Forestry, Chenzhou 423000, China; 3College of Ecology & Environment, Hainan University, Haikou 570228, China; 4Center for Eco-Environmental Restoration Engineering of Hainan Province, Hainan University, Haikou 570228, China; 5Horticulture College, Hunan Agricultural University, Changsha 410128, China; xyyu2011@126.com

**Keywords:** Lavandin, essential oil, Yun-Gui Plateau, monoterpenes, eucalyptol, camphor

## Abstract

Lavandin, as an important cash crop, is cultivated in Kunming, Yun-Gui Plateau of China. For the special growing environment, Lavandin was grown here and used to investigate the changes in the yield and chemical compositions of essential oils extracted from the flowers in different seasons. The essential oils were extracted by hydro-distillation and analysis by gas chromatography-mass spectrometry (GC-MS). Results indicated great changes in chemical composition depending on the season of harvesting. The yields of essential oils ranged from 2.0% to 3.8% among the seasons, and the highest yield was in the summer. Chemical composition data showed that the extracted oils were rich in oxygenated monoterpenes (55.4–81.4%), eucalyptol (38.7–49.8%), camphor (8.41–14.26%), α-bisabolol (6.6–25.5%), and linalool (4.6–12.5%). The contents of eucalyptol and α-bisabolol changed in a contrary trend with seasonal variations. The results provided new insight for Chinese Lavandin germplasm to be used in application and development, and reference to the researcher, the farmer, and investor for sustainable industrialization of the plant grown in the Yun-Gui Plateau of China, but also the similar plateau area of the sustainable developments.

## 1. Introduction

*Lavandula* species are outstanding members of the family Lamiaceae, which are native to the Mediterranean region and south to tropical Africa, with a disjunction to India, and are currently widely cultivated in many regions of the world [1]. The genus is an ornamental and aromatic shrub, which is valuable for the production of essential oils of commercial value as a fragrance, pharmaceutical preparations, and cosmetic products, and is also used in the food industry and ecological agriculture [2,3,4,5,6,7].

These essential oils have been obtained from the flowers, stems, and leaves of the species *L. angustifolia*, *L. hybridia*, and *L. latifolia*, and are classified into three groups on the basis of their content of linalool, linalool acetate, and camphor [8]. Due to their wide economic exploitation, there are many reports on the fragrances of essential oils and identification of constituents from the *Lavandula* species. These studies demonstrate a high degree of intraspecific differences of chemical constituents in the oil, as influenced by genotype, age, development periods, organ, climate, geography, season, and even extraction method, etc. [9,10,11,12,13,14,15,16,17].

Yun-Gui Plateau is a subtropical monsoon climate region with favorable environmental and edaphic conditions for ornamental, aromatic, and medicinal plants. Since the cultivation of *Lavandula* species in Yun-Gui Plateau, Lavandin, blooming in the field throughout the growing season year, and even during winter, has become an important cash crop in the mountainous region. Despite growing interest and the commercial importance of Lavandin, farmers are more interested to imply in the quality and development of essential oil. To the best of our knowledge, there are few reports on the variation of yield and chemical compositions of essential oil from flowers of Lavandin (*Lavandula angustifolia* Mill. × *Lavandula latifolia* Medik.) collected over several months during the flowering season in the plateau of China. The variation in the content of volatile oils related to the flowering phase can help to ensure the required quality and quantity of raw material. This aspect is very important from the point of view of Lavandin for exploitation and utilization. Therefore, the aim of this study was to reveal the variation of yield and chemical composition of Lavandin oils in order to provide a reference for sustainable industrialization of the plant grown in the Yun-Gui Plateau of China, and provided new insight for Chinese Lavandin germplasm to be used in application.

## 2. Results and Discussion

### 2.1. Yields of Essential Oils

The essential oils extracted from the fresh flowers of Lavandin were pale yellow in color with an aromatic-spicy odor. The yields of oils in different seasons are different, where the highest is 90% higher than the lowest (Figure 1), namely, the essential oil yields varied almost two-fold between seasons. The lowest yield (2.0%) was observed in winter, a period with no rainfall. The highest yield (3.8%) was obtained in summer, while the yields ranged from 3.0% to 3.7% in spring and autumn months. The results show that higher temperatures and rainfall favor essential oil biosynthesis during the normal flowering time before harvesting in summer (Figure 1). Furthermore, Pearson correlations showed that the yields of essential oil were significantly positively correlated with temperature (*r* = 0.899, *p* < 0.05), but not significantly positively correlated with precipitation (*r* = 0.812, *p* > 0.05) (Table 1). De Alencar Filho [18] also found that the highest yields occurred in the raining period, but the best essential oil quality of *Croton heliotropiifolius* occurred in the months of lower rainfall, lower relative humidity, and higher temperature, a result similar to in our study. Additionally, a higher yield was obtained from samples collected in spring-B (before) (3.7%) than spring-A (after) (3.0%), and the total rainfall in the period spring-B was higher than spring-A, accounting for 147 and 113 mm, respectively. This may be linked with heavier rainfall and stronger UV radiation in spring and before flowering. Kumari and Agrawal [19] found that the yield of essential oil of Holy basil increased after UV-B treatment. However, Hassiotis et al. [20] found that some rainfall in spring and none immediately before the time of harvesting increased essential oil production. Similarly, previous studies also revealed that differences in the essential oil yields of the other plants could be attributed to seasonal weather variation [21,22,23].

### 2.2. Chemical Compositions of the Essential Oil

Qualitative and quantitative variation in compositions of Lavandin essential oils between seasons were observed, Appendix A report the GC-MS patterns of samples from each season (analysed with an oven temperature program up to 230 °C), and 56, 56, 77, 44, and 64 constituents were identified by GC-MS at the five sampling times. A complete list of identified compounds and their relative contents are summarized in Table 1. Samples taken during different seasons varied in the relative contents of compounds but not in the range of compounds present. This is a new chemotype of *Lavandula* in China, and it was found to contain eucalyptol (38.7–49.8%), camphor (8.4–14.3%), linalool (4.6–12.5%), and α-bisabolol (6.6–25.5%). Other constituents were identified as α-pinene (1.2–3.9%), β-pinene (2.8–7.3%), β-terpieol (1.0–2.2%), terpieol (0.4–1.6%), caryophyllene oxide (0.5–1.3%), and bisabolol oxide B (0.7–1.7%). These essential oils were dominated by oxygenated monoterpenes (55.4–81.4%), followed by oxygenated sesquiterpenes (9.45–29.2%), monoterpene hydrocarbons (5.11–13.5%), and sesquiterpene hydrocarbons (0.21–1.76%) (Table 2, Figure 2).

The seasonal variation showed that most of the main components of these oils reached their peak concentrations in summer. Eucalyptol was the most abundant component observed in all samples, and their contents remained relatively constant (from 45.4% to 49.8%) throughout the growing season. Slight changes were observed in the samples collected in winter and spring-A, with concentrations decreasing by 12–25% (Figure 3). The concentration of camphor varied from 14.3% in autumn to 8.4% in spring-A (Figure 3). Linalool, present at a low concentration (from 8.29% to 4.6%), remained relatively constant from spring-B to spring-A, but increased in summer. The concentration of α-bisabolol was decreased with the decrease of the temperature (Table 2, Figure 3), and reached its maximum value in spring-A (25.5%), almost 4 times the value in spring-B. Our results showed that chemical components of the essential oil of Lavandin may be strongly influenced by several factors, among which the climatic conditions play a major role.

Comparison with previous reports on essential oils in *Lavandula* showed a remarkable difference that seems to mostly depend on plant species and climatic or soil conditions at the growing locations. In the essential oil obtained from different pedo-climatic areas, the main constituents of the most studied species, Lavandin, were linalool, linalyl acetate, and 1,8-cineole (*Lavandula* × *interrnedia* ’Budrovka’, Croatia) [25], linalool, linalyl acetate, camphor, and eucalyptol (*Lavandula × intermedia* Emeric ex Loiseleur, southeast Spain), and [26] 1,8-cineole, borneol, and camphor (*L.× intermedia*, Western Iran) [27]. Nevertheless, the main compounds of Lavandin growing in Kunming of Yun-Gui Plateau of China, eucalyptol, camphor, linalool, and α-bisabolol, had been detected as a new chemotype. As mentioned above, the distribution of essential oil chemotypes was usually concordant with the bioclimatic zones. In this study, the chemotype of Lavandin might be attributed to the climatic conditions of the sampling area, as the Lavandin was growing at the highest altitude (2046 m) and in semiarid conditions. Furthermore, Menary et al. [28] found that the total yield of oils from lavenders (RB, PC, MS, and JP) generally increased over the growing season, and linalool, as the main component, tended to increase with later harvest date, but camphor decreased. The results of this study showed a contrasting change in concentrations of eucalyptol and α-bisabolol over the different flowering seasons.

## 3. Materials and Methods

### 3.1. Collection of Flowers

Lavandin (*L. angustifolia* Mill. × *L. latifolia* Medik.) plants were used in this study. Lavandin cultivars were kindly provided by Dr. Huang (Institute of Urban Environment, Xiamen, China). The inflorescence of Lavandin were collected periodically during the flowering stages in spring (May), summer (August), autumn (October), and winter (December) in order to cover all the different seasons. Lavandin plants were grown at Kunming, Yun-Gui Plateau of China, 25.04° N, 102.73° E, altitude 2046 m, with a plateau monsoon climate and intense ultraviolet radiation, where the daily average value of ultraviolet radiation is 30–59 W/m^2^. The mean annual temperature is 16.0–16.3 °C and mean annual precipitation is 802 mm. In summer, temperature varies from 20.5 to 20.8 °C, while in winter, it varies from 9.5 to 13.1 °C. Fresh flowers were separated from the stem and only flowers were used for the extraction of essential oils.

### 3.2. Sample Preparation

Oils were extracted from the fresh flowers of Lavandin using a Clevenger-type apparatus. A total of 50 g of fresh flower samples and 600 mL of ultrapure water were used, and the hydro-distillation was carried out for 90 min. The procedure was performed in duplicate for different seasons’ samples. Yield of the essential oil was expressed as the mean of two determinations in percentage (V/m). The oil obtained was dried over anhydrous sodium sulfate, filtered, and stored in sealed amber glass vials in a freezer at 4 °C for further analyses.

### 3.3. GC-MS Analysis

The chemical compositions of the extracted oils were analyzed using GC-MS (AGILENT 7890 GC/CMSD 5975) (Agilent, Palo Alto, CA, USA), equipped with a capillary column of HP 5MS (30 m × 250 µm, 0.25 μm film thickness) (Agilent, Santa Clara, CA, USA) and a 70 eV EI Quadrupole detector (Agilent, Santa Clara, CA, USA). Helium was the carrier gas, at a flow rate of 40 mL/min. Injector and ion temperatures were 250 and 230 °C, respectively. The column temperature was initially held at 45 °C for 10 min, then increased by 3 °C/min to 70 °C, and from 70 to 95 °C at 1 °C/min, and increased to 135 °C at a rate of 8 °C/min, and finally increased to 230 °C at 5 °C/min. Diluted samples (1:50 *v*/*v*, in ethylether) of 1.0 μL were injected using a splitless inlet auto sampler. Electron ionization mass spectra were acquired over the mass range 20–500 amu.

### 3.4. Qualitative Analyses

Identifications of components were based on the comparison of retention indices relative to (C_8_–C_20_) n-alkanes with those of the literature and/or with those in close agreement with Reference [29], and further identification was mass spectra of those stored in the spectrometer database using the NIST libraries. All the compounds are expressed in peak area percentage.

### 3.5. Statistical Analysis

The data and plots were analyzed using Origin 2021 (OriginLab, Northampton, MA, USA) and SPSS 23.0 (IBM Corp., Chicago, IL, USA). Yields of essential oils were expressed as means ± standard deviation. Differences were tested with analysis of variance (ANOVA), using the post-test of Student–Newman–Keuls to compare two means. The relationships between yields of the essential oil and climate factors were examined using the non-parametric Pearson’s rank correlation method. All statistical analyses were tested at the 0.05 level of probability.

## 4. Conclusions

Lavandin oil is a popular essential oil which is widely used for many purposes. In the present study, we aimed to create a *L. angustifolia* variety as a cash crop which flowered in the field in all four seasons in the Yun-Gui Plateau. The new chemotype of Lavandin has been studied in China as a source of eucalyptol, linalool, camphor, and α-bisabolol, and which is dominated by oxygenated monoterpenes and oxygenated sesquiterpenes. Moreover, the yields and the main components of essential oils reached the peak production during summer when temperature and precipitation were highest, and seasonal variation in production of eucalyptol and α-bisabolol followed opposite trends. Therefore, the seasonal variation has a great importance in the production of essential oil and influences the quantity and quality of essential oil. This study provides a valid foundation for assessing the quality of Lavandin oil and potential industrial applications. Further study is needed to investigate the effects of growing habitats on the yield, chemical compositions, and antibacterial activity of the essential oil of Lavandin collected in different seasons.

## Figures and Tables

**Figure 1 molecules-26-05639-f001:**
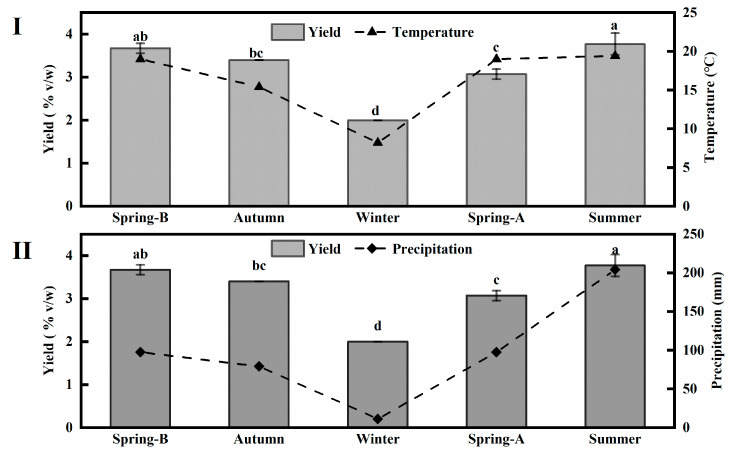
The mean yields of essential oil (%), temperature (°C) (**Ⅰ**), and precipitation (mm) (**Ⅱ**) during the sampling period. Yield defined as the total mass of extracted oils expressed as % of plant material fresh mass; Results are provided as mean ± SD; Values with different letters are significantly different (*p* < 0.05), according to Tukey’s multiple range test; Kunming climate data derived from China Meteorological Administration.

**Figure 2 molecules-26-05639-f002:**
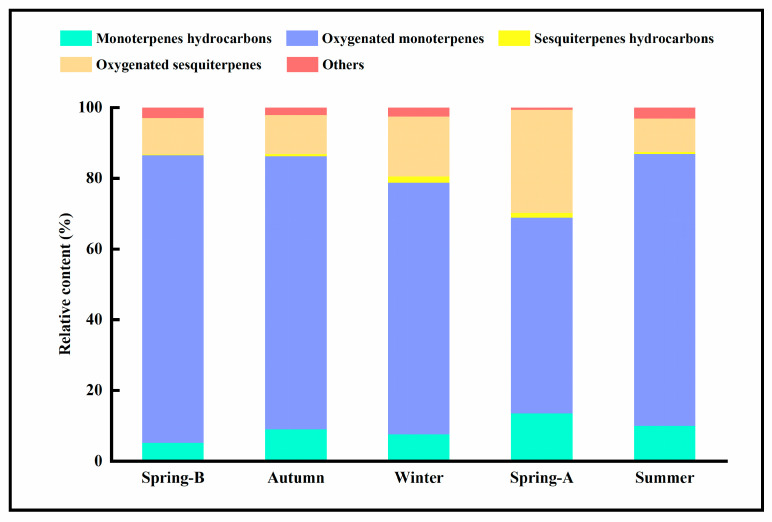
Seasonal variation of main component types and their relative contents in Lavandin essential oil.

**Figure 3 molecules-26-05639-f003:**
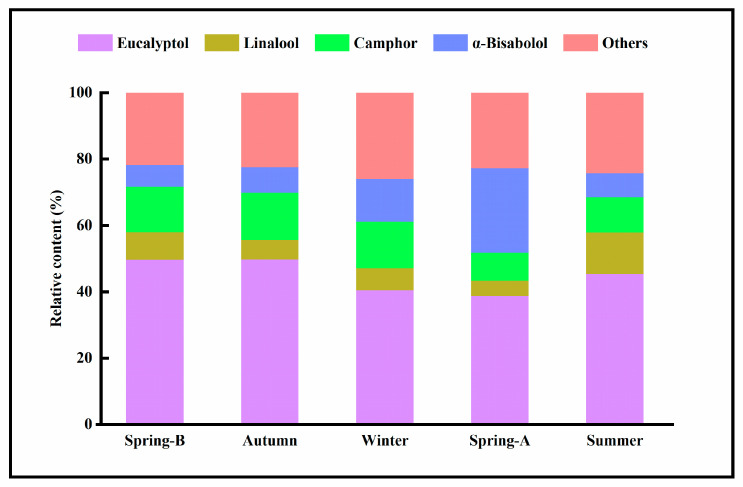
Seasonal variation of the main characteristic components and relative contents in Lavandin essential oils.

**Table 1 molecules-26-05639-t001:** The correlations between yields of essential oil and climate.

Item	Yield	Temperature	Precipitation
Yield	1	0.899 *	0.812
Temperature		1	0.794
Precipitation			1

* Indicates significance at the 0.05 level.

**Table 2 molecules-26-05639-t002:** Essential oil composition of Lavandin isolated from fresh flower mass during different seasons.

No.	RI	Components	Relative Content (%)				
			Spring-B	Autumn	Winter	Spring-A	Summer
1	808	1,3,5-Trioxepane	0.68	0.73	0.63	tr	1.91
2	925	α-pinene	1.15	2.76	1.98	3.87	2.51
3	939	Camphene	0.36	0.60	0.37	0.46	0.44
4	966	β-Pinene	2.82	4.53	3.89	7.33	5.43
5	986	1,2,4,4-Tetramethylcyclopentene	tr	tr	tr	0.12	tr
6	972	1-Octen-3-ol	tr	tr	0.13	tr	0.23
7	981	2,3-Dehydro-1,8-cineole	0.07	0.24	0.15	tr	0.15
8	985	β-Myrcene	0.18	0.21	0.31	0.61	0.49
9	1004	α-Terpinene	0.09	0.14	0.12	tr	0.13
10	1011	o-Cymol	0.42	0.21	0.15	0.05	0.12
11	1018	Eucalyptol	49.70	49.80	40.47	38.72	45.40
12	1028	trans-πOcimene	0.11	0.22	0.57	0.89	0.56
13	1044	γ-Terpinene	0.19	0.26	0.23	0.19	0.22
14	1052	Terpineol, cis-π	tr	0.12	0.20	0.27	0.22
15	1059	cis-Linalool Oxide	0.58	0.31	0.22	tr	0.27
16	1072	Terpinolene	0.08	0.11	tr	0.07	0.14
17	1075	trans-Linalool Oxide	0.65	0.29	0.35	tr	tr
18	1091	Linalool	8.29	5.83	6.60	4.61	12.46
19	1111	1,7,7-Tri methylbicyclo[2.2.1]hept-5-en-2-ol	0.48	0.35	0.32	tr	0.17
20	1119	(+)-Nopinone	0.59	0.31	0.27	tr	0.11
21	1121	(E)-Pinocarveol	tr	tr	0.71	0.31	0.49
22	1126	Camphor	13.64	14.26	14.06	8.41	10.65
23	1128	cis-Verbenol	tr	tr	0.68	0.15	0.44
24	1143	Sabina ketone	0.13	tr	tr	tr	0.05
25	1144	Pinocarvone	0.82	0.77	0.53	0.06	0.32
26	1147	Borneol			0.48	0.15	0.52
27	1151	Terpieol	1.35	1.61	1.36	0.36	1.15
28	1160	Terpinen-4-ol	0.91	0.84	0.70	0.09	0.64
29	1167	Crypton	tr	0.21	0.31	tr	0.15
30	1176	β-Terpieol	0.97	1.41	1.84	2.21	2.18
31	1178	Myrtenal	1.56	tr	0.76	0.08	0.43
32	1177	Myrtenol	tr	tr	0.19	tr	0.35
33	1183	Butanoic acid, hexyl ester	0.49	0.26	0.33	tr	0.22
34	1191	Verbenone	0.80	0.34	0.20	tr	0.14
35	1224	Cuminaldehyde	0.27	0.16	0.20	tr	0.08
36	1227	D-Carvone	0.47	0.37	0.34	tr	0.19
37	1272	Bornyl acetate	0.40	0.23	0.29	tr	0.05
38	1332	Cyclobutanecarboxylic acid, hexyl ester	0.13	0.08	0.06	tr	0.07
39	1371	πMuurolene	tr	tr	0.13	0.15	0.06
40	1378	Geranyl acetate	tr	0.06	0.11	tr	0.05
41	1382	Hexanoic acid, hexyl ester	0.15	0.08	0.15	0.08	0.06
42	1400	1-Caryophyllene	0.12	0.08	0.24	0.35	0.12
43	1423	Π-Bergamotene	tr	0.06	0.13	0.14	0.06
44	1431	π-Sesquiphellandrene	tr	0.06	0.15	0.11	0.05
47	1454	α-Cubebene	tr	tr	0.34	tr	tr
45	1459	trans-p-Mentha-2,8-dienol	0.14	0.28	0.23	tr	0.12
46	1466	πCubebene	tr	0.07	0.09	0.30	0.11
48	1488	Helminthogermacrene	tr	tr	tr	0.13	0.06
49	1497	πMuurolene	tr	0.15	0.24	tr	0.10
51	1570	Caryophyllene oxide	1.27	0.87	1.11	0.66	0.51
52	1599	Cubenol	tr	0.06	0.10	tr	tr
53	1609	Farnesene epoxide, E	0.11	0.09	tr	tr	tr
54	1612	2,2,7,7-Tetra methyltricyclo[6.2.1.0(1,6)]undec-4-en-3-one	0.09	0.10	0.11	tr	tr
55	1630	tau-Cadinol	0.20	0.44	0.67	0.77	0.34
56	1644	(-)-Bisabolol oxide B	1.58	1.70	1.70	1.17	0.71
57	1662	cis-Z-β-Bisabolene epoxide	0.12	tr	tr	tr	tr
58	1673	α-bisabolol	6.62	7.65	12.89	25.53	7.25
59	1728	Santalol, cis, π	tr	tr	tr	tr	0.45
60	1743	Lanceol, cis	tr	tr	tr	0.87	tr
		Monoterpenes hydrocarbons	5.11	8.94	7.54	13.49	9.97
		Oxygenated monoterpenes	81.41	77.28	71.21	55.40	76.85
		Sesquiterpenes hydrocarbons	0.21	0.59	1.76	1.27	0.65
		Oxygenated sesquiterpenes	10.30	11.14	16.94	29.20	9.45
		Total identified	40	56	77	44	64

RI = Retention index (from temperature-programming, using definition of Van Den Dool and Kratz [24]); tr = trace amount (< 0.05%); The bold values indicate the principal compounds present in oils.

## Data Availability

Not applicable.

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
