# Peer review of "Seasonal Variation in Chemical Compositions of Essential Oils Extracted from Lavandin Flowers in the Yun-Gui Plateau of China"

_molecules, 2021, doi:10.3390/molecules26185639_

Round 1
Reviewer 1 Report
This is an interesting study about the composition of lavandin oils and how it changes with seasonal variation. They identified the compounds through NIST libraries and expressed the results as relative concentrations. Considering the large number of compounds detected and the variables studied (seasons, temperature and precipitation), a multivariate analysis would have been more suitable. In addition, I recommend to present the absolute concentration (using external standards) for the main compounds found by GC-MS, such as eucalyptol and camphor, among others.
Author Response
Response to Reviewer 1# of molecules-1348758
Dear Professor Reviewer 1#,
Thank you very much for your insightful comments concerning our manuscript entitled “Seasonal Variation in Chemical Compositions of Essential Oils Extracted from Lavandin Flowers in the Yun-Gui Plateau of China” (molecules-1348758). These comments are highly valuable and very helpful for revising and improving article. We have studied comments carefully and tried our level best to incorporate all the individual points highlighted and revised the whole manuscript for a quality publication. We hope that the quality of this paper is improved and you find our responses to your satisfaction. In the following pages are our point-by-point responses to each of the comments.
Responds to the comments:
- This is an interesting study about the composition of lavandin oils and how it changes with seasonal variation.
Response: Dear reviewer, thank you for appreciation.
- They identified the compounds through NIST libraries and expressed the results as relative concentrations. Considering the large number of compounds detected and the variables studied (seasons, temperature and precipitation), a multivariate analysis would have been more suitable.
Response: Dear reviewer, thank you for valuable advice! With your suggestion that we did a multivariate analysis for the relationships between yields of the essential oil and climate factors (temperature and precipitation), and the correlations were judged in the new Table 1. More details in the first paragraph of Section 2.1.
- In addition, I recommend to present the absolute concentration (using external standards) for the main compounds found by GC-MS, such as eucalyptol and camphor, among others.
Response: Dear reviewer, thank you for valuable advice! In this study, we obtained the relative content of the essential oil components by peak area normalization and expressed in percentages since the unified expression and analysis of all the experimental data.
Blazekovic etc. (2018) and Alencar Filho etc. (2017) reported essential oil of Croton heliotropiifolius, Lavandula x intermedia 'Budrovka' and L. angustifoliam by the relative content of the essential oil components, respectively. At the same time, we used standard substance, i.e., (C8–C20) n-alkanes, eucalyptol, linalool and camphor, for the comparison of retention indices and with NIST libraries to identification of the essential components. We have corrected it with your advice, as follow:
3.4. Qualitative analyses
Identifications of components based on the comparison of retention indices relative to (C8–C20) n-alkanes with those of literature and/or with those in close agreement with references [28], and further identification was mass spectra with those stored in the spectrometer data base using the NIST libraries. All the compounds are expressed in peak area percentage.
The absolute concentration (using external standards) for the main compounds will be used for analysis in our future experimental arrangements. Thank you once more!
References:
Alencar Filho, J. M. T.; Araújo, L. C.; Oliveira, A. P.; Guimarães, A. L.; Pacheco, A. G. M.; Silva, F. S.; Cavalcanti, L. S.; Lucchese, A. M.; Almeida, J. R. G. S.; Araújo, E. C. C., Chemical composition and antibacterial activity of essential oil from leaves of Croton heliotropiifolius in different seasons of the year. Revista Brasileira de Farmacognosia 2017, 27, (4), 440-444.
Blazekovic, B.; Yang, W. F.; Wang, Y.; Li, C.; Kindl, M.; Pepeljnjak, S.; Vladimir-Knezevic, S., Chemical composition, antimicrobial and antioxidant activities of essential oils of Lavandula x intermedia 'Budrovka' and L. angustifolia cultivated in Croatia. Ind Crop Prod 2018, 123, 173-182.
We highly appreciate all your valuable comments and suggestions, and thank you once more!
Kindly regards,
Qing Huang
Reviewer 2 Report
The manuscript presents a theme that has been extensively explored. Based on this fact and for the manuscript to be in a position to be accepted for publication, the following points should be revised:
1) The description of figure legends should be improved, as they are very succinct, and do not express the results placed in the figures;
2) The entire item "Material and Methods" is very succinctly described. There is no citation of references. The topic "Sample preparation" should be rewritten, with details of the extraction process;
3) Present the figures in black and white with clearer captions for the reader;
4) Review the discussion with facts that differentiate the results presented from previously published works.
Author Response
Response to Reviewer 2 # of molecules-1348758
Dear Professor Reviewer 2#,
Thank you very much for your insightful comments, appreciation, your recognition and affirmation to publish in molecular journal concerning our manuscript entitled “Seasonal Variation in Chemical Compositions of Essential Oils Extracted from Lavandin Flowers in the Yun-Gui Plateau of China” (molecules-1348758). Those comments are well valuable and very helpful for revising and improving our article, as well as the important guiding significance to our researches. In the following pages are our point-by-point responses to each of the comments. We hope meet with approval and thank you in advance!
Responds to the comments:
- The manuscript presents a theme that has been extensively explored. Based on this fact and for the manuscript to be in a position to be accepted for publication
Response: Dear reviewer, thank you for positive comments, appreciation and your recognition and affirmation to publish in molecular journal !
The following points should be revised:
1) The description of figure legends should be improved, as they are very succinct, and do not express the results placed in the figures;
Response: Dear reviewer, thank you for instructive suggestions, we have modified according to your suggestion. To make the figure legends clearer for the reader to understand, All of them modified as follow:
Figure 1. The mean yields of essential oil (%), temperature (℃) (Ⅰ) and precipitation (mm) (Ⅱ) during the sampling period”.
Figure 2. Seasonal variation of main component types and their relative contents in Lavandin essential oil”.
“Figure 3. Seasonal variation of the main characteristic components and relative contents in Lavandin essential oils”.
- The entire item "Material and Methods" is very succinctly described. There is no citation of references. The topic "Sample preparation" should be rewritten, with details of the extraction process;
Response: Dear reviewer,thank you for valuable suggestions! We have revised the section as follow:
3.2. Sample preparation
Oils were extracted from the fresh flowers of Lavandin using a Clevenger-type apparatus. A total of 50 g of fresh flowers samples and 600 ml of ultrapure water were used, and the hydrodistillation was carried out for 90 min. The procedure was done in duplicate for different seasons samples. Yield of the essential oil was expressed as the mean of two determinations in percentage (V/m). The oil obtained was dried over anhydrous sodium sulfate, filtered and stored in sealed amber glass vials in a freezer at 4℃ for further analyses.
3.3. GC-MS analysis
The chemical compositions of the extracted oils were analyzed using GC-MS (AGILENT 7890 GC/CMSD 5975) equipped with a capillary column of HP 5MS (30 m × 250 µm, 0.25 μm film thickness) and a 70-eV EI Quadruapole detector. Helium was the carrier gas, at a flow rate of 40 ml/min. Injector and ion temperatures were 250 and 230℃, respectively. The column temperature was initially held at 45℃ for 10 min, then increased by 3℃/min to 70℃,and from 70℃ to 95℃ at 1℃/min, and increased to 135℃ at a rate of 8℃/min, and finally increased to 230℃ at 5℃/min. Diluted samples (1:50 v/v, in ethylether) of 1.0 μL were injected using a splitless inlet auto sampler. Electron ionization mass spectra were acquired over the mass range 20 ~ 500 amu.
3.4. Qualitative analyses
Identifications of components based on the comparison of retention indices relative to (C8–C20) n-alkanes with those of literature and/or with those in close agreement with references [28], and further identification was mass spectra with those stored in the spectrometer data base using the NIST libraries. All the compounds are expressed in peak area percentage.
3.5. Statistical analysis
The data were analysed using Origin 2021 (OriginLab, USA) and SPSS 23.0 (IBM Corp., USA). Yields of essential oils were expressed as means ± standard deviation. Differences were tested with analysis of variance (ANOVA), using the post-test of Student–Newman–Keuls to compare two means. The relationships between yields of the essential oil and climate factors were examined using non-parametric Pearson’s rank correlation method. All statistical analyses were tested at 0.05 level of probability.
Reference:
Adams, R.P., Identification of essential oil components by gas chromatography/mass spectrometry, 4th Ed. Allured Publishing Corporation, Carol Stream, IL. USA, 2007.
- Present the figures in black and white with clearer captions for the reader;
Response: Dear reviewer, thank you for valuable advice! We use the colourful figures which would be more clearly since in modern publish online.
4) Review the discussion with facts that differentiate the results presented from previously published works.
Response: Dear reviewer, thank you for instructive suggestions! A proper sentence was added to summarize the third paragraph of the discussion as “Our results showed that chemical components of the essential oil of Lavandin may be are strongly influenced by several factors among which the climatic conditions play a major role”.
The fourth paragraph of the discussion was update as “Nevertheless the main compounds of Lavandin growing in Kunming of Yun-Gui Plateau of China, eucalyptol, camphor, linalool and α-bisabolol, had been detected as a new chemotype. As mentioned above, the distribution of essential oil chemotypes was usually concordant with the bioclimatic zones. In this study, the chemotype of Lavandin might be attributed to the climatic conditions of the sampling area, as the Lavandin was growing at the highest altitude (2046 m) and with semiarid conditions. Furthermore, Menary et al. [27] found the total yield of oils from lavenders (RB, PC, MS and JP) generally increased over the growing season, and linalool, as the main components, tended to increase with later harvest date, but camphor decreased. While the results of this study show a contrary change in concentrations of eucalyptol and α-bisabolol over the different flowering season.”
In a word, thank you once more, and appreciate your positive, constructive comments and advice the molecular to accept our manuscript.
Kindly regards,
Qing Huang
Reviewer 3 Report
The manuscript is well written and the topic and results are well related to the aims and scopes of the journal. However, I consider that in its current version and with the results provided, it does not provide enough scientific advance to be published in this prestigious journal.
The introduction is too short, and taking into account that there is talk of a single product in a given region, its socio-economic importance could have been highlighted. Furthermore, it should be specified whether similar studies have been carried out and their importance further emphasized. I doubt that the data provided in this version are of great interest to readers of the journal, and of little practical application. Also, I do not consider the relationships between meteorology/rainfall and oil production. I think that a lot more experimental data should be used so that a relationship could really be established, if any. The conclusions provide virtually nothing, not even a summary of the study. Although I agree that more experiments are needed.
Author Response
Response to Reviewer 3 # of molecules-1348758
Dear Professor Reviewer 3 #,
Thank you very much for your instructive comments concerning our manuscript entitled “Seasonal Variation in Chemical Compositions of Essential Oils Extracted from Lavandin Flowers in the Yun-Gui Plateau of China” (molecules-1348758). Those comments are well valuable and very helpful for revising and improving our article, as well as the important guiding significance to our researches. In the following pages are our point-by-point responses to each of the comments. We hope meet with approval and thank you in advance!
Responds to the comments:
1) The manuscript is well written and the topic and results are well related to the aims and scopes of the journal.
Response: Dear reviewer, thank you for positive comments and appreciation!
- However, I consider that in its current version and with the results provided, it does not provide enough scientific advance to be published in this prestigious journal.
Response: Dear reviewer, thank you for instructive comments. And we sure will pay more attention in the future our study. At the same time, we believe the findings of this study can provide reference for our peers, especially for local industrial development. Anyway, thank you for costing time and energy on our manuscript.
- The introduction is too short, and taking into account that there is talk of a single product in a given region, its socio-economic importance could have been highlighted. Furthermore, it should be specified whether similar studies have been carried out and their importance further emphasized. I doubt that the data provided in this version are of great interest to readers of the journal, and of little practical application.
Response: Dear reviewer, thank you for instructive comments. We have modified as follow:
Yun-Gui Plateau is a subtropical monsoon climate region with favourable environmental and edaphic conditions for ornamental, aromatic and medicinal plants. Since the cultivation of Lavandula species in Yun-Gui Plateau, Lavandin, blooming in the field throughout the growing season year, and even during winter, have become an important cash crop in the mountainous region. Inspite of growing interest and the commercial importance of Lavandin, farmers are more interested to imply in the quality and development of essential oil. To the best of our knowledge, there are few reports on variation of yield and chemical compositions of essential oil from flowers of Lavandin (Lavandula angustifolia Mill. × Lavandula latifolia Medik.) collected over several months during the flowering season in plateau of China. The variation in the content of volatile oils related to the flowering phase can help to ensure the required quality and quantity of raw material. This aspect is very important from the point of view of Lavandin for exploitation and utilization. Therefore, the aim of this study was to reveal the variation of yield and chemical composition of Lavandin oils in order to provide a reference for sustainable industrialization of the plant grown in the Yun-Gui Plateau of China, and provided new insight for Chinese Lavandin germplasm to be used in application.
- Also, I do not consider the relationships between meteorology/rainfall and oil production. I think that a lot more experimental data should be used so that a relationship could really be established, if any.
Response: In order to judge the relationships between yields of the essential oil and climate factors (temperature and precipitation), we used non-parametric Pearson’s rank correlation method for a multivariate analysis, the result showed in the first paragraph of Section 2.1 that “the yields of essential oil were significantly positively correlated with temperature (r = 0.899, p <0.05), but no significantly positively correlated with precipitation (r = 0.812, p >0.05) (Table 1)”.
- The conclusions provide virtually nothing, not even a summary of the study. Although I agree that more experiments are needed.
Response: Dear reviewer, thank you for instructive comments. We have modified as follow with your comments.
Lavandin oil is a popular essential oil which is used widespread for many purposes. With the present study we targeted to create a L. angustifolia variety as a cash crop which however flowered in the field all over the four seasons in the Yun-Gui Plateau. The new chemotype of Lavandin has been studied in China as a source of eucalyptol, linalool, camphor, α-bisabolol, and which is dominated by oxygenated monoterpenes and oxygenated sesquiterpenes. Moreover, the yields and the main components of essential oils reached the peak production during summer when temperature and precipitation were highest, and seasonal variation in production of eucalyptol and α-bisabolol followed opposite trends. Therefore, the seasonal variation has a great importance in the production of essential oil and influenced on the quantity and quality of essential oil. This study provides a valid foundation for assessing the quality of Lavandin oil and potential industrial applications. Further study is needed to investigate the effects of growing habitats on the yield, chemical compositions and antibacterial activity of the essential oil of Lavandin collected in different seasons.”
Anyway, sincerely thank you for your kindness instructive comments, and costing time and energy on our manuscript!
Best wishes!
Qing Huang
Round 2
Reviewer 1 Report
Regarding multivariate analysis, I think a principal components analysis (PCA) or partial least squares discriminant analysis (PLS-DA) would be more suitable than Pearson's correlation.
Reviewer 2 Report
Highlighted points have been corrected or improved, resulting in better manuscript quality.